# Evaluation of the Effect of Waste from Agricultural Production on the Properties of Flexible Polyurethane Foams

**DOI:** 10.3390/polym15173529

**Published:** 2023-08-24

**Authors:** Joanna Paciorek-Sadowska, Marcin Borowicz, Marek Isbrandt

**Affiliations:** Department of Chemistry and Technology of Polyurethanes, Faculty of Materials Engineering, Kazimierz Wielki University, J. K. Chodkiewicza Street 30, 85-064 Bydgoszcz, Poland; m.borowicz@ukw.edu.pl (M.B.); m.isbrandt@ukw.edu.pl (M.I.)

**Keywords:** flexible polyurethane foams, bio-filler, polyurethane properties, corncake

## Abstract

The management of by-products and waste from agriculture and the agri-food industry is a challenge for the plastics industry. Flexible polyurethane foams (EPPUR) containing ground corncake from corn oil production were obtained. The influence of the bio-filler on the physico-mechanical and thermal properties of synthesized flexible polyurethane foams was investigated. The content of corncake ranged from 0 php (part per 100 parts of polyol) to 10 php. Open-cell flexible polyurethane foams with a favorable comfort factor were obtained. The lower reactivity of the developed polyurethane systems was conducive to the formation of cells of slightly smaller sizes but of a more regular shape in comparison with the foam not modified with the bio-filler. Measurements of the mechanical properties indicated that the modified foams held had similar or even better properties than the reference sample without bio-filler.

## 1. Introduction

Scientific research in the field of polymers has been the driving force behind human progress for over 100 years, and the development of the plastics industry has become one of the phenomena of modern times [1,2,3,4,5]. Currently, research on the synthesis of new polymeric materials is a “gateway to the future” because it provides increasingly unique materials for the needs of all fields of technology and economy. They focus mainly on broadly understood issues related to the basics of chemistry, technology, processing and recycling processes, and the analysis of their functional properties determining applications in many industries [6,7].

Plastics are an inexhaustible source of new innovations contributing to sustainable development, increasing safety, and improving the quality of life [8,9,10,11,12]. For many years, we have observed an upward trend in their production. The plastics industry is one of the most important areas of the modern world economy. Over the last few decades, these materials have rapidly displaced the traditional, hitherto used materials from many industries [13,14]. The expansion of plastics is justified primarily in the economy and in the wide and constantly developing range of desirable and beneficial properties of polymers. The market is dominated mainly by the production of polypropylene, polyethylene, poly(vinyl chloride), and polystyrene [15,16]. Polyurethanes are also at the forefront of the most popular materials. Their production and use increases on average by over 3% annually. Undoubtedly, the versatile properties of polyurethanes contribute to the dynamics of growth and forecasts of further development, which have been maintained for several years. They can be freely modified in a very wide range, e.g., as a result of changes in the chemical structure or the addition of modifiers, obtaining materials for many applications [17,18,19,20,21,22,23,24,25,26,27,28,29]. Depending on the composition of the formulation and processing conditions of polyurethanes, a wide range of products is obtained: foams, rubbers, varnishes, adhesives, fibers, and leather-like materials [30,31,32]. A special position among polyurethane materials is occupied by flexible polyurethane foams (FPUf), which are light, durable materials with excellent functional properties. Thanks to the possibility of designing the properties of materials at the stage of developing the formulations, polyurethane foams find many applications. The demand for products made of flexible polyurethane foam generates a lot of industrial sectors, such as furniture, automotive, footwear, textile, machinery, and household appliances production. However, in order to obtain materials with a wide range of applications, it is necessary to comprehensively take into account all aspects of the process, starting from the synthesis of materials, their processing capabilities, and the final physical and mechanical properties of the obtained product.

The issue of selecting the parameters of the production process is not simple because there are a number of factors affecting this process and the quality of the obtained product [33,34]. Currently, the producers of polyurethane foams face new challenges, which are based on meeting not only high technical requirements but also ecological aspects [35]. The development of technology for the production of polyurethane foams is aimed at introducing into them raw materials from renewable sources, such as, for example, vegetable oils [35,36,37,38,39,40,41,42]. Renewable raw materials have the potential to make a huge difference in chemical synthesis, including the synthesis of flexible polyurethane foams. Currently, the trend of searching for technologies for obtaining polyurethane bio-composites is getting stronger.

Modification of conventional polyurethane materials with additives in the form of bio-fillers is dictated by the reduction in production costs and the improvement of functional properties. The use of these environmentally friendly, sustainable materials also effectively contributes to reducing greenhouse gas emissions and promotes environmental protection. Important from the point of view of environmental protection is the fact that very often bio-fillers are waste products. Among the renewable materials currently used in the polyurethane industry, vegetable fillers have gained much interest [43,44].

It is well known that fillers improve the density and mechanical, optical, electrical, and thermal properties of polymers. In addition, they positively affect the economic aspect of the production of composites, reducing the price of the final product. Numerous studies suggest that the addition of natural fibers to composite polyurethane foams improves the mechanical properties of such materials [45,46].

This research is aimed at developing and optimizing formulations for obtaining new composite materials based on flexible polyurethane foams with the use of corncake as a filler and determining the relationship between the percentage of bio-filler and the internal structure, functional and mechanical properties of the polyurethane composite material. The main aim of the research was to optimize the content of ground corncake in the flexible polyurethane foam matrix in order to obtain composite materials with better properties. Corncake is a by-product of the production of corn oils. The scale of this issue is evidenced by the fact that the weight ratio of the amount of cake to oil obtained from corn seeds is, on average, 70:30. This means that from 100 kg of corn seeds, about 70 kg of cake and 30 kg of corn oil are obtained.

Corn is an industrial grain that is a rich source of carbohydrates, especially starch. The world harvest of corn grain has long exceeded a billion tons. During the last two decades, corn production has doubled. The use of corn processing products, including oils, also increased. Corn oil belongs to the group of oils with increased nutritional value due to the high content of unsaturated acids, among which linoleic acid has the largest share.

The oilcake obtained during oil production contains anti-nutritive substances (glucosinolates, phytic acid, and its derivatives), which negatively affect the growth and metabolism of animals during feeding. In addition, very often, during inappropriate storage conditions (e.g., too high humidity and temperature as well as long storage time), cakes spoil. Therefore, excessive feeding of oilcakes to animals is not recommended [47]. Another way to manage corncakes is to use them as biofuels for heating purposes. However, they are not a good fuel due to their high content of protein and amino acids. During combustion, these compounds generate tarry substances that emit nitrogen oxides and sulfur oxides, which in turn are unfavorable for the environment and contribute to the corrosion of heating system pipes.

The growing demands of consumers and the tightening of international standards force producers of all industries to constantly modernize technology, search for innovative solutions, and enter the market with new, more ecological products. The development of an innovative method of utilizing corncakes as a bio-filler in the production of polyurethanes meets the problems of their overproduction.

## 2. Materials and Methods

### 2.1. Materials

In order to obtain physically modified flexible polyurethane foams (FPUf) with the assumed functional properties, the type, size, and proportions of the reinforcing material, the structure of the polyurethane matrix, and the production technology should be taken into account. Rokopol F3600 polyether polyol, a block-statistical copolymer of ethylene oxide and propylene oxide based on glycerin, was used for the synthesis of FPUf. It was in the form of a homogeneous, clear liquid that contained antioxidants (without BHT). The polyol has a hydroxyl number of 48 mg KOH/g and a dynamic viscosity of 580 mPa·s. It has been supplied by PCC Rokita SA (Brzeg Dolny, Poland). Ongronat 1080 was used as the isocyanate raw material. It was a mixture of isomers of 2,4- and 2,6-toluene diisocyanate. It had a viscosity of 3 mPa·s (at 25 °C) and a content of 2,4-isomers of 80%. It has been supplied by BorsodChem (Wanhua, China).

The following raw materials were used as additives: Tegostab BF 2370 (Evonik Industries AG, Essen, Germany) as a silicone-based surfactant; DABCO 33LV (Evonik Industries AG, Essen, Germany) as an amine catalyst to activate the gelation reaction between the isocyanate and the polyol; DABCO BL-11 (Evonik Industries AG, Essen, Germany) as an amine catalyst activating the reaction between water and isocyanate; Kosmos T-9 (Evonik Industries AG, Essen, Germany) tin octoate, organometallic catalyst activating the gelation reaction. Water was used as a chemical blowing agent necessary to generate carbon dioxide in situ.

Corncake was used as a physical modifier of polyurethane foams. It was supplied by Agrovet (Śniadowo, Poland). These cakes were supplied in the form of pellets, which were ground in a laboratory mill. A powder with a grain diameter of less than 1 mm was obtained. The detailed share of individual fractions determined using the sieve analysis method is presented in Table 1. The chemical composition of corncakes depends on the method of oil pressing and the species and genetic features of the corn. The physical form of corncake and an example of flexible polyurethane foam modified using ground corncake are shown in Figure 1. The chemical composition of the ground bio-filler is shown in Table 2 (data obtained from the supplier).

### 2.2. Obtaining Flexible Polyurethane Foams Containing Ground Corncake

The process of foaming polyurethane foams includes several stages, the course of which has a significant impact on the cellular structure and properties of the produced foam materials. These stages are characterized using such parameters as time, free-rise rate, reaction temperature, and dielectric polarization. The production of flexible polyurethane foams was carried out on a laboratory scale using a FOAMAT device (Format Messtechnik GmbH, Karlsruhe, Germany). FPUf modified with corncake was obtained using a one-step method in a two-component system (A and B). During laboratory work, the mixture of component A (polyol raw material and additives) and component B (diisocyanate) was prepared in separate cups. For this purpose, the appropriate amount of component A was weighed into a suitable cup, and then the carefully calculated amount of component B according to the prepared formulation. Component A was obtained using carefully weighing and mixing appropriate amounts (based on the previously prepared formulation—Table 3): petrochemical polyol, catalysts, surfactant, blowing agent, and ground corncake. Component B was pure diisocyanate. One reference foam (REF) was obtained, which did not contain bio-filler, and five foams were modified with corncake, in which the cake content increased from 2 php up to 10 php (every 2 php in each subsequent foam).

The polyol premix is prepared to thoroughly mix the non-reactive components involved in the process. This provided quick and short contact of the reacting components with each other and, at the same time, the course of appropriate reactions. In addition, it was easier to regulate the dosing of ingredients. The entire mixture was quickly and thoroughly mixed (mixing time—10 s, stirrer rotation—1800 rpm) in the appropriate mass ratio and left to rise freely. The time of foam rise was observed, paying particular attention to the moment when its rise was completed. After the rise phase, the foam was degassed. During this time, the foam cells ruptured, releasing the gas inside and giving the foam an open-cell structure. The phenomenon of a slight fall of the foam and a decrease in its height by 2–5 mm was observed. After rising the foam in the mold and measuring the technological times, the foam was placed in a circulating dryer at a temperature of 120 °C for about 2 min to harden its structure. After removing the foam from the dryer and a positive result of the organoleptic assessment, the foam was left for 72 h for seasoning. Each foam was prepared three times. After seasoning, the obtained foams were cut into samples of appropriate size and tested for performance properties according to the standards described in the next section.

### 2.3. Methods

#### 2.3.1. Analysis of Foaming Process

The use of fillers in the polyurethane formulation requires a very thorough study of the process of obtaining foams and their functional properties. First of all, using natural fillers may lead to a change in the proportions between urethane and urea bonds, a change in the cross-linking density, and a change in the structure of the matrix. As a consequence, an unfavorable effect of such a modification on the mechanical properties of the obtained composites can be observed.

The analysis of the FPUf foaming process was carried out using a FOAMAT (Format Messtechnik GmbH, Karlsruhe, Germany) device equipped with specialized Format-Messtechnik software FOAM 4.0 (Format Messtechnik GmbH, Karlsruhe, Germany). The apparatus enabled the performance of reproducible measurements as a function of time of such parameters as foam rise rate, pressure, process temperature (T), and dielectric polarization (D).

#### 2.3.2. Scanning Electron Microscopy (SEM) Examination

The structure of the foams was analyzed using a Hitachi SU8010 Scanning Electron Microscope (SEM) from Hitachi High-Technologies Co. (Tokyo, Japan). The tests were carried out at an accelerating voltage in the range of 10–30 kV, at a working distance of 10 mm and magnification of 130×. 

#### 2.3.3. Fourier Transform Infrared (FTIR) Spectroscopy Examination

The presence of characteristic groups of FPUfs was identified using infrared spectroscopy (FTIR). FTIR spectra were taken with a Nicolet iS20 spectrophotometer (Thermo Fisher Scientific, Waltham, MA, USA) in the range of 400 to 4000 cm^−1^.

#### 2.3.4. Physico-Mechanical Tests

The examination of the external appearance was carried out immediately after the polyurethane foaming process and after its thermostating. The foam rise process was evaluated. Visual evaluation of the foam included the appearance of the foam surface (presence of large bubbles and cracks, shrinkage, and discoloration). These characteristics indicate whether all the ingredients have reacted with each other or have been properly mixed.

Determination of the apparent density was carried out according to PN-EN ISO 845:2010 from a sample thermostated before determination for 2 h at a temperature of 40 °C and subjected to air conditioning. The apparent density was calculated as the ratio of the mass of the material to its total geometric volume, i.e., to the space occupied by the sample, including the gas-filled cells.

Determination of hardness (compressive stress, CLD_40_) was performed according to PN-EN ISO 3386-1:2000 on a Zwick/Roell Z005 universal testing machine (Ulm, Germany). Hardness is the value of pressure needed to compress the foam by 40% in relation to the initial height of the foam. The designation expressed the load capacity of the foam.

Rebound elasticity was determined according to DIN 53573 using a Schöb elastomer type 5109 device (Gibitre Instruments, Bergamo, Italy). A pendulum with a potential energy of 0.196 J with a 101 g mass attached to it was used for the tests. A foam sample of (80 ± 2) mm × (80 ± 2) mm and (50 ± 2) mm height was placed in a device attached to the apparatus and struck with a weight attached to a pendulum. The arithmetic mean of the results of five tests of the same sample was to be taken as the determination result.

The comfort factor (SAG factor) was calculated based on ISO 3386-1. This parameter was calculated as the ratio of the force needed to compress the foam by 65% to the force needed to compress the foam by 25%.

Permanent deformation was determined according to PN-EN ISO 1856:2004/A1:2008 on a universal testing machine. The principle of the test consisted of keeping the tested sample for a certain time, at a certain temperature, under a constant load and noting the thickness of the sample before the test (initial thickness) and after removing the load (final thickness). 

Determination of resistance to repeated compression was performed according to PN-EN ISO 3385:1999. The step height of the compression plate for the test sample was 75% of the initial thickness. The test was also performed using the Zwick/Roell Z005 universal testing machine by using the measurement procedures provided with the TestExpert III software (Zwick/Roell, Ulm, Germany). The resistance to repeated compression concerned two parameters—the loss of foam thickness (Δh) and the loss of its hardness (ΔCLD_40_). Thickness loss after repeated compression was determined from Equation (1).
(1)$$\Delta h=\frac{h_{0}-h}{h_{0}}\cdot 100\%$$

Δh—loss of thickness after repeated compression [%];

h_0_—initial thickness of foam [mm];

h—thickness of foam after repeated compression [mm].

Hardness loss after repeated compression was determined from Equation (2).
(2)$$\Delta CLD_{40}=\frac{iCLD_{40}-eCLD_{40}}{iCLD_{40}}\cdot 100\%$$

ΔCLD_40_—loss of hardness after repeated compression [%];

iCLD_40_—initial hardness of foam [kPa];

eCLD_40_—hardness of foam after repeated compression [kPa].

## 3. Results and Discussion

### 3.1. Foaming Process

Polyurethane foams are made of three-dimensional structures, forming gas-filled cells, separated from each other by thin walls forming a polyurethane matrix. Depending on the type and amount of raw materials used to obtain foams, they may have a different cellular structure, including the size and number of cells and the thickness of the cell walls. This directly affects the performance properties of the obtained polyurethane foams [48,49,50]. The structure of the polyurethane matrix depends not only on the designed formulation but also on the conditions of the foaming process [51]. A number of different processes take place during the preparation of flexible foams. These processes should be sufficiently synchronized with each other so that the resulting foam has an appropriate matrix structure for the required functionality. Changes in such parameters as foam rise time, reaction temperature, and dielectric polarization during the process of obtaining reference foam and foams modified the smallest and largest amount of corncake are presented in Figure 2.

After mixing the components, a rapid increase in temperature in the system was observed because the polyurethane foam synthesis reaction is an exothermic reaction. The addition of a bio-filler in the form of corncake into the system resulted in changes in the foaming parameters. Due to the fact that it was assumed to test the effect of the filler on the FPUfs foaming process, the polyurethane formulation was based on the identical qualitative and quantitative selection of basic raw materials. After mixing the premix with the bio-filler, an increase in the viscosity of the mixture was observed. At the same time, the reactivity of the entire system was reduced. This is a natural phenomenon when using physical fillers, which has been observed in many research studies [52,53,54]. The analysis of the FOAMAT results showed that the REF system was characterized by the highest reactivity, and its changes can be observed on the basis of the course of the dielectric polarization curve. This parameter referred directly to the processes and reactions that took place during the transition of the liquid mixture of polyol raw material, diisocyanate, and additives to the form of cured foam. From the value of dielectric polarization, we can conclude about the degree of conversion of substrates. The change of the dielectric polarization curve occurred the fastest for the REF system. It reached the value of 0 after 186 s. The analysis of the course of the remaining curves indicated a much slower course of the foaming reaction in the system containing ground corncake. This was confirmed using a slight decrease in the dielectric polarization value. The dielectric polarization for the CC2% foam reached the value of 0 after 210 s, while for the CC10% foam, it reached the value of 0 after 235 s. This also confirmed the longer gelation times of foams modified with bio-filler. This was most likely due to the reduced expansion of the cells formed during the foaming process and the formation of additional nucleation centers, which formed more cells during the reaction. A similar dependence was observed by Bartczak et al. They used coffee grounds and sawdust as bio-fillers in a polyurethane formulation [55]. The highest temperature value was obtained for the REF foam, in which the core temperature reached 131 °C within 209 s. It was observed that the amount of bio-filler added into the system reduced the maximum temperature during the foaming process compared to the reference foam. The addition of 2 php of corncake decreased the core temperature to 120 °C. In turn, with its highest content, the maximum foam core temperature of 115 °C was recorded in time of 238 s. It can be seen that the dominant reactions in the first phase of the process were coming to an end. This was evidenced by the inflection point of the temperature-time curve and the decrease in dielectric polarization. The process of foaming flexible polyurethane foams modified with corncake was characterized using different dynamics of the course. It was disturbed by the higher viscosity of the polyol premix and its lower homogeneity (presence of bio-filler particles). This affected both the reaction temperature and the rate of dielectric polarization decrease.

### 3.2. Structure and FTIR Analysis of the Obtained FPUfs

Many functional properties of flexible polyurethane foams depend on the proportion of open cells. The theory is that most of the cell opening occurs when the foam reaches its largest cell size. At this time, the polymer is characterized using a very high viscosity and very low extensibility. High viscosity causes a problem with the rapid movement of the structural elements of the emerging polymer and, as a result, prevents the release of the increasing gas pressure inside the cells. Ultimately, cells burst, and an open-cell structure is formed, characteristic of flexible polyurethane foams [56,57]. FPUf, which has an open cell structure, is characterized using increased elasticity and flexibility, which results in extending its useful life. However, open-cell polyurethane foam allows air to circulate, so using it, e.g., in a mattress, allows for an appropriate level of ventilation. The cellular structure of the obtained flexible polyurethane foams was assessed on the basis of micrographs taken using scanning electron microscope (Figure 3a–d).

The SEM micrographs showed that the addition of bio-filler to the polyurethane matrix caused the reduction of cells present in the structure of the reference foam. The added bio-filler particles acted as nuclei of forming cells, which resulted in the formation of a larger number of them. The addition of a powder filler can change the nucleation from homogeneous to heterogeneous and reduce the nucleation energy. This favors the formation of small cells with a more rounded shape. The cells of the unmodified REF foam had a tetrahedral structure, which was the result of the lower viscosity of the REF system. In addition, the ground corncake increased the viscosity of the reaction mixture, which inhibited the coalescence of the formed gas bubbles [58]. It can be observed that the filler was built into the cell walls, and additional small cells were formed near its particles, which supported the “flexibility” of the polyurethane matrix. The formation of additional cells can be associated with the release of water bound in the structure of bio-filler particles and reacting with isocyanate groups. Water bound to the filler was released at the elevated temperature reached by the reaction mixture when its viscosity increased as a result of the reaction of the polyol with the diisocyanate.

FTIR spectral analysis was carried out for all the obtained flexible polyurethane foams. The summary of the FTIR spectra is shown in Figure 4.

The presence of characteristic absorption bands for polyurethane foams was noted on the FTIR spectra of the obtained foams: 3290–3300 cm^−1^ bands belonging to valence vibrations of N-H bonds of urethane and urea groups; 2866–2970 cm^−1^ bands belonging to the valence vibrations of C-H bonds of alkyl groups; 1720–1745 cm^−1^ bands belonging to the valence vibrations of carbonyl groups C=O of urethane and urea groups; 1630–1640 cm^−1^ band belonging to the amide groups; 1598 cm^−1^ band belonging to the C=C double bonds present in the aromatic ring derived from the diisocyanate feedstock; the bands in the ranges of 1530–1540 cm^−1^, 1449–1453 cm^−1^ and 1406–1409 cm^−1^ come from the structures of aromatic rings characteristic of diisocyanates used in the synthesis of flexible polyurethane foams; 1372–1374 cm^−1^ and 1097–1098 cm^−1^ bands belonging to the C-C bond in the polyurethane structure; 1224–1225 cm^−1^ band belonging to the C-O-C ether bond in the chain derived from the polyether polyol; 922–925 cm^−1^ band belonging to C-H bond in aliphatic chains and 690–700 cm^−1^ band belonging to the C-H bond in aromatic rings. 

### 3.3. Physico-Mechanical Properties of FPUfs

The properties of flexible foams depend on the method of foaming, the type of raw materials used for their production, the chemical structure, and the cellular structure of the polyurethane matrix. By selecting the composition of the formulation during the foaming process, it is possible to obtain FPUfs characterized by various physical and mechanical properties, such as apparent density, rebound flexibility, SAG factor, hardness (CLD_40_), tensile strength, elongation at break or resistance to repeated compression.

The development of the FPUf formulation requires a detailed analysis of the obtained results. This is due to the need to select the proportions between the formulation components that enable obtaining a product with assumed properties that meet the application criteria. An important role in the large variety of applications is played by the possibility of modifying the physical and mechanical properties as well as the functional properties of polyurethane materials. Wide application possibilities force the production of a large and diverse range of FPUfs. One of the most important criteria for the selection of foam for specific applications is its apparent density. The apparent density of the foam affects its durability. Low-density foam is less resistant to oxidation and embossing degradation. Low-density foam is less resistant to oxidation degradation and kneading. Choosing the right apparent foam density is a compromise between durability and price. The lower the apparent density, the lower the price and, at the same time, the lower the durability of the foam. It is assumed that the optimal apparent density of good quality flexible foam should be in the range of 25–30 kg/m^3^.

The addition of ground corncake to the polyurethane formulation contributed to the reduction of the apparent density of the obtained foams from 30.4 kg/m^3^ for the REF foam (without the bio-filler) to 27.1 kg/m^3^ for the CC10% foam, containing 10% of the bio-filler. It is clearly visible that the apparent densities of the obtained foams were in the range of the apparent density values of classical FPUfs for many applications. The reason for the decrease in the apparent density of the foams containing ground corncake was the water contained in it, in the amount of about 5% of the weight of the added bio-filler. Water contained in organic fillers can be a problem for most thermoplastics because it can initiate hydrolytic degradation. As a result, deterioration of the functional properties of such materials is observed. In the case of polyurethane foams, the additional water content added into the system with the organic filler will not adversely affect the performance parameters of the foams because it will be used for reactions with the diisocyanate, creating a blowing agent—carbon dioxide. A higher amount of blowing agent reduced the apparent density of the materials. In addition, the organic filler added into the polyurethane system most often acts as a nucleating agent to promote the growth process to rapidly form a low-density foam. A similar dependence was observed by Zhang et al., who used bio-fillers in the form of pine bark and walnut shells [59].

The apparent density of FPUfs was closely related to their rebound flexibility (resilience). The dependence of both parameters on the content of corncake in foams is shown in Figure 5.

On the basis of the course of the curves, it was clearly visible that with the decrease in the apparent density of FPUfs, their rebound flexibility increased. The REF foam had the lowest value of this parameter. It amounted to 43.32%. It is worth noting that this foam was also characterized by the highest apparent density (30.4 kg/m^3^). The CC10% foam, with the highest content of corncake and the lowest apparent density, was characterized by the highest rebound flexibility. It was 46.12%. In addition to the apparent density and rebound flexibility of FPUfs, very important parameters determining the application possibilities of these materials are the comfort factor (SAG factor) and foam hardness (CLD_40_). Figure 6 shows the dependence of the SAG factor and the hardness of the obtained foams, depending on the amount of used bio-filler.

The CLD_40_ hardness of standard flexible polyurethane foams is within a relatively wide range of this parameter, i.e., 1.5–4.6 kPa. The obtained foams containing corncake fell within this range. On the other hand, for foams used for the production of seats and mattresses, the SAG factor should be between 2.0 and 3.0 [60]. All obtained flexible foams modified with bio-filler had this parameter within the assumed range. However, it should be remembered that the SAG factor should be selected individually. This is due to the fact that when calculating the SAG factor according to the standard, a certain thickness of the foam and its deformation percentage is assumed. For this reason, it is important to select the right material for a specific product and user requirements. The CLD_40_ hardness values of the obtained foams modified with the bio-filler slightly differed from each other. Their values were close to 4 kPa. These small differences had a decreasing trend with the increasing amount of bio-filler. They were caused by a change in the structure of the obtained foams and the incorporation of corncake into the walls of the forming cells. Most likely, this resulted in a slight weakening of the polyurethane matrix by reducing hardness.

The addition of a bio-filler in the form of ground corncake into the FPUfs matrix contributed to a favorable decrease in permanent deformations in comparison with the reference foam. This dependence is shown in Figure 7.

After the tests, a slight decrease in permanent deformation was observed by increasing the amount of ground corncake. FPUfs with bio-filler were characterized by lower permanent deformation (value 4.10% for the foam with the highest content of corncake) compared to the reference foam (value 5.35%). It was observed that the higher the elasticity of the foams, the lower the permanent deformations of the foams. The decrease in permanent deformations can usually be the result of an increase in the number of soft segments in materials with a filler caused by an increased amount of water added with the bio-filler into the formulation. During the foaming of foams with bio-filler, a decrease in the reaction temperature of their preparation was observed in comparison with the reference foam. Such a change indicated that the introduction of ground corncake slowed down the reaction rate between the –OH and –NCO groups. This resulted in the formation of more urea groups in the soft segments during foaming.

Tensile strength and elongation at break are parameters whose values depend on the structure of the polyurethane matrix but also on the water content in the formulation [60]. Figure 8 shows the results of these parameters depending on the amount of added corncake.

Observing the changes in the tensile strength of the obtained flexible foams, it can be seen that the CC2% foam modified with the smallest amount of corncake was characterized by the highest value of this parameter, amounting to 121.0 kPa. Increasing the content of bio-filler in the formulation resulted in foams with a gradual decrease in tensile strength up to 101.00 kPa. The reference foam had a value of this parameter of 99.0 kPa. In contrast, the elongation at break increased with increasing amount of bio-filler in the formulation from 104% for the REF foam to 141% for the CC10% foam with the highest filler content. These dependencies were caused by the formation of a hard urea segment as the amount of water in the system (added with the bio-filler) increased. In addition, both parameters were also affected by the weaker dispersion of bio-filler particles in systems with a higher share of bio-filler. Uneven distribution of the filler in the polyurethane matrix may favor the formation of small defects in the structure, which in turn can reduce the tensile strength at large fillings.

An important performance parameter of flexible polyurethane foams is their resistance to repeated compression, expressed as loss of thickness and loss of hardness. These parameters define, e.g., the durability of foams during their everyday use. The dependence of these parameters on the bio-filler content is shown in Figure 9.

It is usually very difficult to reduce the hardness value and, thus, to obtain greater flexibility in the case of polyurethane foams. Controlling the amount of filler added in the polyurethane formulation may lead to a decrease in foam hardness, but also very often leads to undesirable changes in other physical and mechanical properties of these materials. When testing the resistance to repeated compression of the obtained flexible foams, it was observed that the loss of thickness of the CC10% foam containing 10 php of bio-filler was lower than the loss of thickness of the reference foam by about 19%. However, in the case of loss of hardness after repeated compression of the obtained samples, it can be stated that the changes were insignificant and amounted to about 1% in the comparison of the reference foam and the foam with the highest content of bio-filler.

It can be concluded that despite the decrease in the apparent density of foams with corncake, no unfavorable changes in strength were noted after the repeated compression test. The corncake used in the polyurethane formulation makes it possible to produce a more homogeneous and ordered foam structure. In addition, this structure was strengthened by the formation of more urea groups as a result of increasing water content in FPUf formulation (water contained in corncake).

## 4. Conclusions

As part of the conducted research, a formulation and a method for obtaining flexible polyurethane foams modified with a bio-filler in the form of corncake were developed. The research focused on checking the possibility of using different amounts of bio-filler and its effect on the performance properties of FPUfs. The use of corncake in the polyurethane system allowed us to obtain flexible foams with more favorable properties than foams available on the market. However, the increase in the content of bio-filler in foams caused changes in the foaming process, e.g., reducing the reactivity of the system, lowering its temperature and dielectric polarization, and increasing the viscosity of the premixes. In addition, a significant reduction in the apparent density, hardness, and favorable reductions in permanent deformations of the modified materials were observed in comparison with the values of these parameters corresponding to the reference foam without bio-filler. Foams modified with corncake showed favorable changes in performance parameters, which are more comfortable to use, e.g., in the form of a mattress made of such material, and have a longer time of use.

Our research has contributed to a better understanding of the influence of bio-fillers on the structure and properties of FPUfs and the comparison of modified and unmodified materials. The ecological aspect of the works carried out cannot be overlooked. They included the use of waste material and, at the same time, obtaining more environmentally friendly polymeric materials.

## Figures and Tables

**Figure 1 polymers-15-03529-f001:**
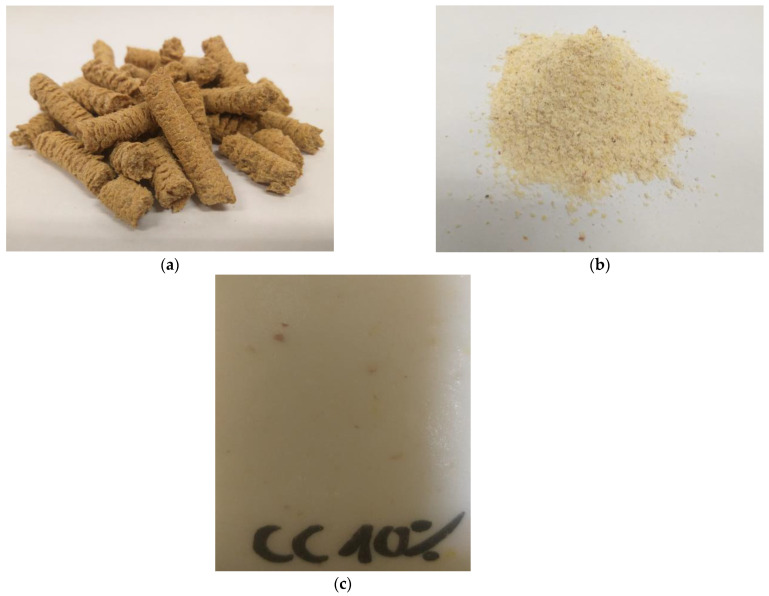
The appearance of corn pellets (**a**), ground cake (**b**), and flexible polyurethane foam containing 10 php of corncake (**c**).

**Figure 2 polymers-15-03529-f002:**
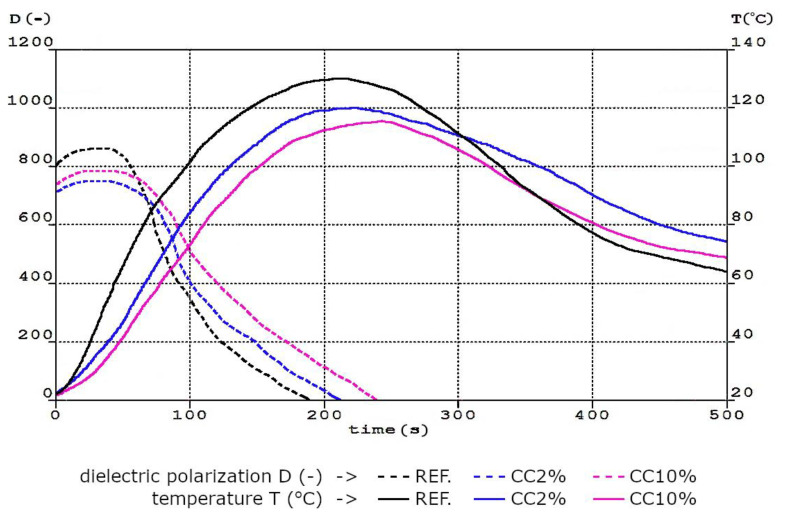
Dependence between temperature (T), reaction time and dielectric polarization (D) of the reaction mixture during the foaming process for REF foam and CC2% and CC10% foams.

**Figure 3 polymers-15-03529-f003:**
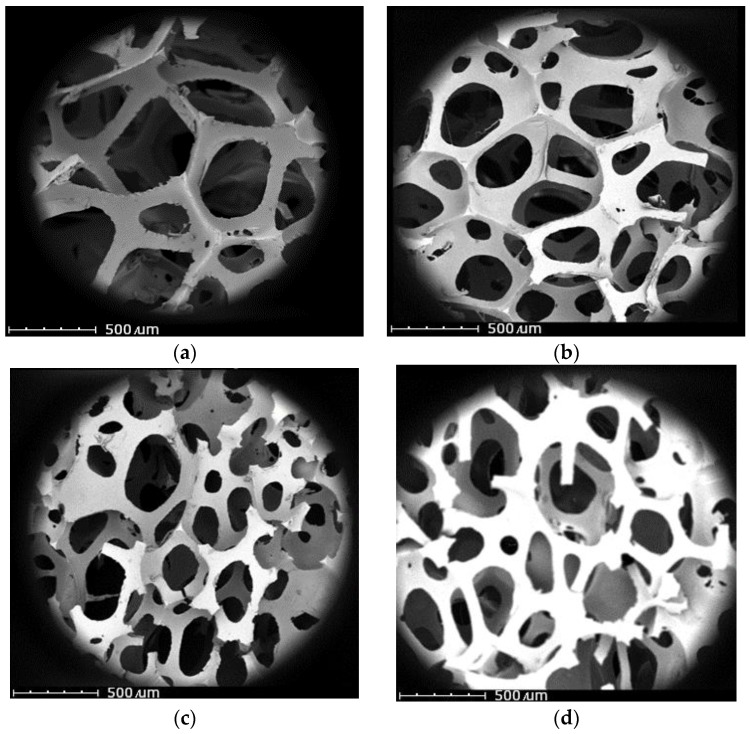
SEM micrographs of the cellular structure of the reference foam REF (**a**) and modified foams CC2% (**b**), CC6%, (**c**) CC10% (**d**).

**Figure 4 polymers-15-03529-f004:**
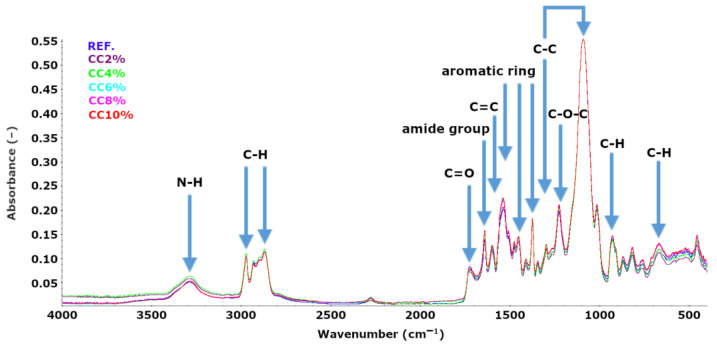
Comparison of FTIR spectra of REF foam and foams modified with corncake.

**Figure 5 polymers-15-03529-f005:**
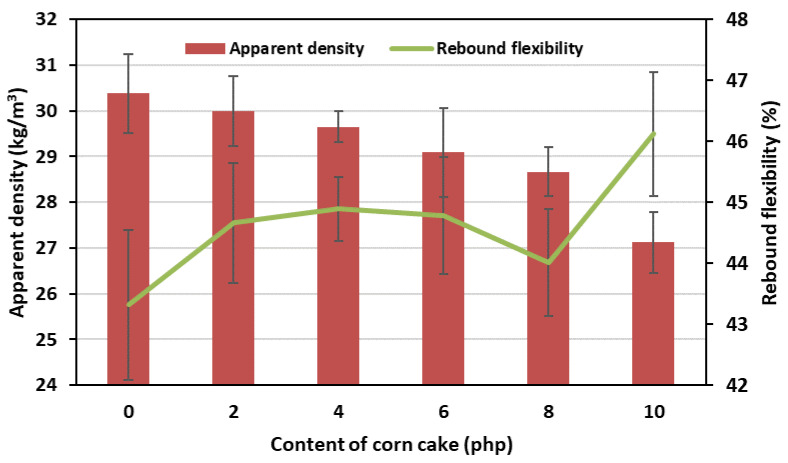
Dependence of the apparent density and rebound flexibility on the content of corncake in flexible foams.

**Figure 6 polymers-15-03529-f006:**
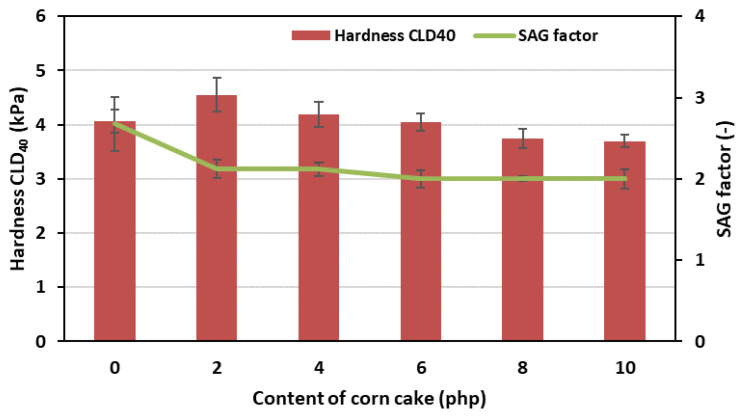
Dependence of the hardness CLD_40_ and SAG factor on the content of corncake in flexible foams.

**Figure 7 polymers-15-03529-f007:**
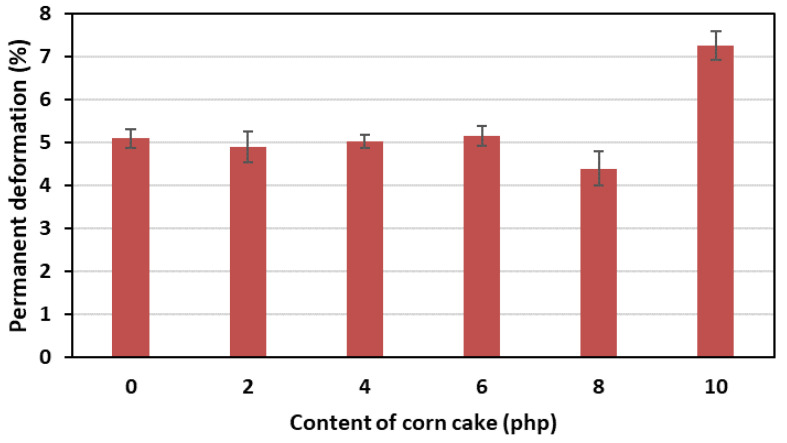
Dependence of permanent deformation on the content of corncake in flexible foams.

**Figure 8 polymers-15-03529-f008:**
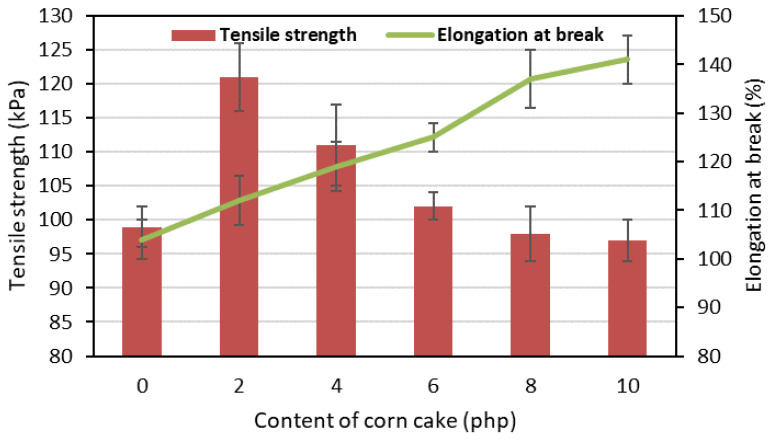
Dependence of the tensile strength and elongation at the break on the content of corncake in flexible foams.

**Figure 9 polymers-15-03529-f009:**
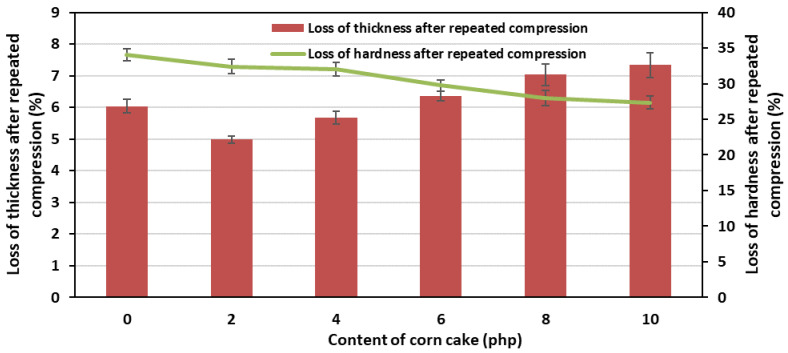
Dependence of loss of thickness and loss of hardness after repeated compression on the content of corncake in flexible foams.

**Table 1 polymers-15-03529-t001:** Results of sieve analysis of ground corncake.

Fraction on the Sieve	Percentage Content (%)
800–1000 µm	3.45
600–800 µm	11.87
400–600 µm	14.89
200–400 µm	65.46
100–200 µm	2.42
50–100 µm	0.87
<50 µm	1.04

**Table 2 polymers-15-03529-t002:** Chemical composition of corncake (in 1 kg of cake).

Component	Dry Mass (%)	Water Content (%)	Ash (%)	Proteins (%)	Fat (%)	Raw Fiber (%)	Calcium (%)	Phosphorus (%)
Content	95.00	4.70	9.00	36.50	4.20	14.20	0.33	0.85

**Table 3 polymers-15-03529-t003:** Formulation of flexible polyurethane foams modified using corncake.

Foam	Rokopol F3000 (g)	DABCO 33LV (g)	DABCO BL-11 (g)	TEGOSTAB BF 2370 (g)	KOSMOS T-9 (g)	Distilled Water (g)	Corncake *(php)* (g)	Ongronat 1080 (g)
**REF**	100.00	0.20	0.05	1.00	0.20	4.00	*0* 0.00	47.41
**CC2%**	100.00	0.20	0.05	1.00	0.20	4.00	*2* 2.00	47.41
**CC4%**	100.00	0.20	0.05	1.00	0.20	4.00	*4* 4.00	47.41
**CC6%**	100.00	0.20	0.05	1.00	0.20	4.00	*6* 6.00	47.41
**CC8%**	100.00	0.20	0.05	1.00	0.20	4.00	*8* 8.00	47.41
**CC10%**	100.00	0.20	0.05	1.00	0.20	4.00	*10* 10.00	47.41

## Data Availability

Not applicable.

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
