# Peer review of "Evaluation of the Effect of Waste from Agricultural Production on the Properties of Flexible Polyurethane Foams"

_polymers, 2023, doi:10.3390/polym15173529_

Round 1

Reviewer 1 Report

This manuscript is well organized in all aspects of research background & purpose, experimental method and results, and conclusion, so can be adopted by Polymers; however, additional explanations are required for the following prior to publication.

1.  Data for morphology and particle size distribution of corn powder fillers are required. The surface characteristics of the filler and the size and shape of the filler affect the formation of air cells in the foam structure.

2. Qualitative information can be obtained through SEM images on cell size, but it is difficult to accurately consider changes in particle size or size distribution.

Reviewer 2 Report

This article is well written and the present is sound and clear. The reviewer suggest some additional polish to make it better:

1. Label your FTIF with functional group location can help reader understand easily.

2. Scale bar in microscopy images should be larger. Current one is impossible to read.

3. A picture as Figure. 1 to graphically illustrate your entire procedures, which make this article more attractive and easier to catch the content.

4. Any pictures of how the foam looks like? I would suggest a photo in Figure. 1(c)

5. Separate the testing method from sample preparing method. Testing method is better in group, like: mechanical testing, microscopy, spectroscopy....
